# Biocompatibility and Hemolytic Activity Studies of Synthesized Alginate-Based Polyurethanes

**DOI:** 10.3390/polym14102091

**Published:** 2022-05-20

**Authors:** Kashif Zafar, Khalid Mahmood Zia, Rami M. Alzhrani, Atiah H. Almalki, Sameer Alshehri

**Affiliations:** 1Department of Applied Chemistry, Government College University, Faisalabad 38030, Pakistan; kashifzafar@gcuf.edu.pk; 2Department of Chemistry, Government College University, Faisalabad 38030, Pakistan; 3Department of Pharmaceutics and Industrial Pharmacy, College of Pharmacy, Taif University, P.O. Box 11099, Taif 21944, Saudi Arabia; r.zhrani@tu.edu.sa (R.M.A.); s.alshehri@tu.edu.sa (S.A.); 4Department of Pharmaceutical Chemistry, Taif University, P.O. Box 11099, Taif 21944, Saudi Arabia; ahalmalki@tu.edu.sa; 5Addiction and Neuroscience Research Unit, College of Pharmacy, Taif University, Al-Hawiah, Taif 21944, Saudi Arabia

**Keywords:** biopolymer, polyurethane composites, alginate

## Abstract

Many investigators have focused on the development of biocompatible polyurethanes by chemical reaction of functional groups contained in a spacer and introduced in the PU backbone or by a grafting method on graft polymerization of functional groups. In this study, alginate-based polyurethane (PU) composites were synthesized via step-growth polymerization by the reaction of hydroxyl-terminated polybutadiene (HTPB) and hexamethylene diisocyanate (HMDI). The polymer chains were further extended with blends of 1,4-butanediol (1,4-BDO) and alginate (ALG) with different mole ratios. The structures of the prepared PU samples were elucidated with FTIR and ^1^H NMR spectroscopy. The crystallinity of the prepared samples was evaluated with the help of X-ray diffraction (XRD). The XRD results reveal that the crystallinity of the PU samples increases when the concentration of alginate increases. Thermogravimetric (TGA) results show that samples containing a higher amount of alginate possess higher thermal stability. ALG-based PU composite samples show more biocompatibility and less hemolytic activity. Mechanical properties, contact angle, and water absorption (%) were also greatly affected.

## 1. Introduction

Biocompatibility and biodegradability are important properties of polymers that have recently gained immense attention due to applications in various surgical and medical fields [1]. PU is a synthetic polymer, embracing biocompatible characteristics. PU is synthesized via a condensation reaction of diol and diisocyanate [2]. Bioactivity in the material can be improved by the utilization of biomolecules such as chitosan [3] and alginate. Because of this ability, one can synthesize a large number of composite materials of PU with desired properties such as biodegradability, biocompatibility, thermal stability, and durability [4]. PU products range from coatings to elastomers, foams, and adhesives. PU has many alternating soft and hard segments. The soft segment consists of a polyol and hard segments consist of diisocyanate and a chain extender [5]. Due to blood compatibility, PU has lots of applications in surgical appliances, heart pacemakers, catheters, transplants for joints, and sutures [6,7]. Several human body implants such as dental materials and bone-fixing materials have steadily improved body performance for the long-term [8]. In the recent past, progress in controlled drug delivery [9], gene therapy, regenerative medicine, and tissue engineering have demanded the need for some novel properties of bio-composites, such as biodegradability [10,11]. In this regard, biologically derived natural biomolecules and artificially synthesized biodegradable polymers have attained significant attention [12]. Biopolymers with diverse properties are required for a combination of vital properties, such as biodegradability and biocompatibility in synthetic polymers like PU [13]. Alginate (C_6_H_8_O_6_)_n_ is a natural biocompatible polysaccharide extracted from brown algae. Due to its diverse biomedical applications, it has become useful in medical science [9,14,15]. Alginate dressings are used for wound healing [15]. It also prevents or reduces the risk of wound tissues bacterial infection. Additionally, it supports the proliferation of cells that helps in the fast recovery of injured tissue and minimizes the probability of inflammation [16]. Inflammation happens mostly due to the presence of free radicals. The reason for the reduced risk of inflammation during the process of wound healing is due to the antioxidant property of alginate [13]. Recent studies showed a number of advantages of synthetic biopolymers (alginate) including the potential to develop a sustainable industry as well as enhancements in various properties, such as biocompatibility, biodegradability, durability, flexibility, high gloss, clarity, and tensile strength. The biochemical properties of alginate, such as antioxidant, anti-infectious, and anti-inflammatory properties result in the wound-healing processes. Alginate boosts the process of wound healing through the attachment in tissue remodeling, collagen deposition, and granulation tissue formation. Recent research has revealed that alginate also boosts the process of epithelial recovery, vascular density, and fibroblast proliferation. The introduction of polysaccharides, e.g., alginate as chain extenders or cross-links for a part of the hard segment can enable us to obtain non-toxic, enhanced biocompatible, and mechanical properties of PU products [17]. Keeping in view the available reported literature, there is a dire need to synthesize a polysaccharide-based biodegradable PU with the potential to have biodegradability, biocompatibility, and bio-functionality. In previous studies, a theoretical approach reporting recent advances and perspectives on alginate-based PUs [17] and the in vivo antidiabetic and wound-healing potential of chitosan-, alginate-, maltodextrin-, or Pluronic-based mixed polymeric micelles has been investigated [18]. In this continuation, alginate and calix [4] arenes-modified graphene oxide nanocomposite beads and alginate-based bionanocomposites have also been reported [19].

From the last few decades, the trend of the utilization of polysaccharides in various industrial fields is rapidly increasing, owing to their structural diversity, biodegradability, biocompatibility, abundance, non-toxicity, and specific bioactive properties. The most abundant marine polysaccharide, alginate, with its inherent well-known gelling and stabilizing properties, proved to be a potential candidate for synthetic modified biomaterials [17].

Sodium alginate, procured from Sigma-Aldrich, is the sodium salt of alginic acid, a natural polysaccharide found in brown algae. Keeping in view the versatile properties and applications of the cell wall components of marine brown algae and due to the absence of any report on the synthesis and characterization of alginate-based biocompatible PUs with detailed investigations including FTIR, NMR, XRD, DSC, TG analyses, mechanical and swelling properties, contact angle measurements, and toxicity levels of PU composites are discussed and interpreted in this paper. The preparation of alginate-based PU composites were designed to improve the physicochemical properties of PU materials compared to other available materials [20].

## 2. Experimental Procedure

### 2.1. Chemicals

Dimethyl formamide (DMF ≥ 99.0%), 1,4-butanediol (1,4-BDO ≥ 99.0%), hydroxyl-terminated polybutadiene (HTPB ≥ 99.0%), hexamethylene diisocyanate (HMDI ≥ 98%), dimethyl sulfoxide (DMSO ≥ 99.0%) and alginate (ALG ≥ 99.0%) were purchased from Sigma-Aldrich Chemicals (St. Louis, MO, USA). All chemicals were of analytical grade.

#### Alginate (ALG ≥ 99.0%)-Preliminary Information

Sodium alginate is a kind of polysaccharide extracted from kelp-like Phaeophyceae, formed by α-l-mannuronic acid (M section) and β-d-guluronic acid (G section) and connected through 1,4-glucosidic bond. Sodium alginate is a cell wall component of marine brown algae and contains approximately 30 to 60% alginic acid. It appears as a white or light yellow powder, which is odorless and tasteless. Its molecular formula is (C_6_H_7_O_6_Na)_n_. The viscosity of alginate solutions increase as pH decreases and reaches a maximum around pH = 3.0–3.5 as carboxylate groups in the alginate backbone become protonated and form hydrogen bonds. Alginic acid is used as a hydrocolloid in various applications, such as food manufacturing, pharmaceuticals, and textiles and cosmetics, particularly as an emulsifier, and is also used in dentistry to make molds.

### 2.2. Synthesis of PU Materials

PU was synthesized by condensation polymerization of hexamethylene diisocyanate (HMDI) and hydroxyl-terminated polybutadiene (HTPB) [21,22]. The sample PUK-1 was prepared in two steps; in the first step, the PU-prepolymer was synthesized by the reaction of HTPB and HMDI with a mole ratio of 1:3, respectively, under a blanket of inert media of dry N_2_, and the reaction temperature was maintained at 100 °C. The progress of the reaction was monitored by measuring –NCO contents using FTIR and the back titration method. After the preparation of the prepolymer, the PU was further extended by chain extender (1,4-BDO). The 1,4-BDO was added to the reaction mixture with vigorous stirring to get final the PU. After the completion of the reaction, the trapped moisture was evolved by keeping the polymer sample in a vacuum prior to casting. The polymer was cured in an oven with a hot-air-circulating heating system for 24 h. Cured samples of polymers were kept for one week prior to testing at room temperature (25 °C) and a humidity level of 40%. Figure 1a shows systematic explanation for the preparation of PUs.

For the synthesis of PUK-2 to PUK-4, prepolymer was synthesized by following the same route as discussed in the previous section (PUK-1). Then, a combination of the chain extenders ALG and 1,4-BDO were added with the mole ratio as mentioned in Table 1. For this purpose, alginate procured from Sigma-Aldrich was used after depolymerization with a treatment of hydrogen peroxide. ALG was dissolved in the DMSO and introduced into the prepolymer at 70 °C. The reaction was continued for 10 min till homogeneity and high viscosity was achieved. The prepared polymer sample was removed from the reaction flask, placed in a vacuum for 15 min, and cured at 100 °C for 24 h. 

For the synthesis of PUK-5, prepolymer was synthesized by following the same route as discussed in the above sample (PUK-1). Then, the chain extender alginate was added as a substitute of 1,4-BDO. Alginate was dissolved in the DMSO, then introduced into the prepolymer at 70 °C. The reaction was continued for 10 min till homogeneity and high viscosity were achieved. The prepared polymer sample was removed from the reaction flask, placed in a vacuum for 15 min, and cured at 100 °C for 24 h.

### 2.3. Measurements

Fourier-transform infrared spectroscopy (FTIR) scans of the samples were collected on completely dried thin film cast on KBr (potassium bromide) discs from a dimethylformamide (DMF) solution in absorbance mode. Infrared measurements were performed on a Bruker (Ettlingen, Germany) IFS 48 FTIR spectrometer. The ^1^H-NMR spectra was recorded in a deuterated dimethyl sulfoxide (DMSO-d6) solution with a Bruker Advance 400-MHz spectrometer.

Chemical shift (d) values were given in parts per million with tetramethylsilane as a standard, following a reported method [18]. X-ray diffractograms of the polymers were obtained in a Siemens D-5000 diffractometer with radiation Cu-Ka (λ = 15.4 nm, 40 Kv, and 30 mA) at 25 °C. The relative intensity was registered in a dispersion range (2θ) of 5–70°. Measurements of the weight loss of prepared samples were recorded on TGA Analyzer under the N_2_ atmosphere from room temperature up to 600 °C with a heating rate of 20 °C/min. Differential scanning calorimetry (DSC) was carried out at a heating rate of 10 °C/min using a DSC-8000 (Perkin Elmer London, UK) under the inert environment. Hemolytic activity of PU materials was tested following a standard reported procedure [19,20].

Mechanical properties such as tensile strength and elongation at break were recorded on an Instron Mechanical Testing instrument by using stress–strain curves at a strain rate of 50 mm/min [23]. All measurements were carried out at 25 °C on 1 mm thick polymer film according to ASTM412.

## 3. Results and Discussion

### 3.1. FTIR Analysis

The spectrum of HMDI was taken afresh, interpreted, and compared with the previous study [24], which represents all the characteristic peaks of diisocyanate. In the spectrum of HMDI, there are two peaks observed at 2866 cm^−1^ and 2941 cm^−1^, attributed to the CH_2_ symmetric and anti-symmetric stretching vibrations respectively. It also exhibited a very sharp peak at 2264 cm^−1^ on account of the –NCO functional group present in HMDI; this peak helps us to study the extent of the reaction and it is very prominent before the chain-extending reaction. The intensity of this peak gradually decreased as the reaction proceeded and diminished in the final PU composite product after the chain-extending reaction.

The FTIR spectrum of hydroxyl-terminated polybutadiene was also taken afresh and compared with the previous study [25]. An FTIR scan of HTPB showed all the characteristic peaks of polyol. A broad peak was observed at 3340 cm^−1^, ascribed to the hydrogen bonding of –OH groups and it is noticeable in the spectrum of HTPB [21]. This peak almost disappeared in the spectrum of prepolymer due to the complete reaction of –OH groups with –NCO groups of the HMDI. The reason for the consumption of all –OH groups is that HTPB was taken in a lesser amount (1 mole) in the reaction as compared to diisocyanate (3 moles). The bands observed at 2614 cm^−1^ and 2540 cm^−1^ were attributed to the antisymmetric and symmetric –CH_2_ stretching vibrations, respectively. A small band observed at 1436 cm^−1^ is ascribed to the stretching vibrations of aliphatic dienes –C=C–. The presence of these peaks confirmed the structure of unsaturated polyol [22].

The FTIR spectrum of 1,4-butanediol was compared with that which was previously reported [25]. The FTIR spectrum represented all the characteristic peaks of 1,4-BDO. A prominent peak was observed at 3300 cm^−1^, attributed to –OH stretching. Two other peaks at 2935 and 2870 cm^−1^ were also observed as being due to the CH_2_ antisymmetric and symmetric stretching vibrations [4,23].

The FTIR spectrum of NCO-terminated PU prepolymer [26] exhibited a small band at 3344 cm^−1^, attributed to the stretching of the –NH group that corresponds to the presence of hydrogen bonding in the prepolymer [27,28]. The band at 1555cm^−1^ is attributed to the presence of a –NH group. Due to a –OH group of HTPB, the peak at 3409 cm^−1^ disappeared, justifying that –OH groups reacted with –NCO to form urethane NH linkage, which confirmed the formation of the PU prepolymer [29]. A peak at 1730 cm^−1^ is ascribed to the (stretching) of the carbonyl group. The appearance of a new peak –NH (urethane group) and a decrease in the peaks of HMDI and –NCO proved the formation of prepolymer due to the reaction of polyol and diisocyanate. Other peaks in the prepolymer spectra are: 1597 cm^−1^ (C–O group hydrogen-bonded stretching), 1727cm^−1^ (carbonyl stretching), 2866 cm^−1^ (CH_2_ stretching vibration), 1450 cm^−1^ (–CH_2_ bending vibration), 1414 cm^−1^ (CH bending), and 1531 cm^−1^ (–NH deformation). However, the appearance of a –NH peak at 3339 cm^−1^, the disappearance of a –OH peak at 3344 cm^−1^ and a decrease in a –NCO peak at 2249 cm^−1^, proved the formation of the PU prepolymer.

The FTIR spectrum of the alginate- and 1,4-BDO (chain extenders)-based PU is depicted in Figure 2. The alginate- and 1,4-BDO-based PU was synthesized as a result of the reaction between a –NCO-terminated prepolymer with chain extenders [30]. Synthesized composites displayed a prominent wide vibration band at 3000–3500 cm^−1^ whereas, in the blended samples, the carbonyl band shifted to a significantly lower position near wave number 1701 cm^−1^. The presence of alginate in the synthesized polymer composite material was confirmed due to the uronic band of alginate in the blend sample. In addition to these, other peaks were also observed in the region from 2600–2800 cm^−1^, attributed to antisymmetric and symmetric CH_2_ stretching. The presence of peaks attributed to urethane linkage confirmed the reaction of the prepolymer and chain extender. All the samples showed characteristic peaks of urethane functional groups.

### 3.2. ^1^H NMR Studies of PUs

Much research has been performed using ^1^H NMR spectroscopy to confirm the structure of alginate-based PUs [25]. The ^1^H NMR spectra of the pristine 1,4-BDO PUs, 1,4-BDO- and ALG blend-based PU, and pristine ALG-based PUs are shown in Figure 3a, 3b, and 3c, respectively. In all the PU samples, the signals of methylene protons appeared at 4.94 ppm (H5), 1.2 ppm (H4), 2.0 ppm (H3), 5.34 ppm (H2), and 3.94 ppm (H1). The ^1^H NMR spectra of PUK-1 is shown in Figure 3a. The signal observed at 7.5 ppm is attributed to –NH groups of urethane linkages and the protons of CH_2_ groups of 1,4-BDO showed a peak of 3.4 ppm (H9) [26,27,28]. The ^1^H NMR spectra of the 1,4-BDO- and alginate-based PU (PUK-3) is shown in Figure 3b. The spectrum of PUK-3 showed all the signals of 1,4-BDO at the proposed positions [29,30]. The peak of 5.05 ppm is ascribed to the anomeric protons of α-L-G. The peak of 4.8 ppm is ascribed to the anomeric proton of β-D-M and H5 of α-L-G adjacent to the β-D-M [31]. Similarly, a peak in the anomeric hydrogen (M4) of alginate attached to the urethane group appeared at 3.3 ppm, adjacent to the peak of the proton (H9) of 1,4-BDO (3.4 ppm). Both peaks of the protons of 1,4-BDO and alginate are almost in equal in the sample in which equal amounts of chain extenders were added. The ^1^H NMR spectrum of pristine ALG-based PU (PUK-5) was in conformity with the proposed structure. The spectrum of PUK-5 showed all the signals of ALG at the proposed positions [31]. It is apparent from Figure 3 that the peak of the 1,4-BDO proton diminished and the peak intensity of the anomeric proton (M4) of alginate increased. These signals prove the synthesis of ALG-based PU and are encouraging for further studies.

### 3.3. XRD Analysis of Prepared Composites

The crystallinity of the prepared samples was observed by using X-ray diffraction (XRD) analysis. The diffractograms of the prepared samples are displayed in Figure 4 and XRD data is given in Table 2. The observed 2θ value for PUK-1 was 24° with d-spacing at 3.71 Å and peak intensity at 20,100 a.u. In sample PUK-2, a blend of chain extenders 1,4-BDO and alginate were used with a ratio of 1.5:0.5 moles. The angle 2θ for this sample was observed at 24.7 with a peak intensity of 20,500 a.u. and d-spacing at 3.60 Å. These results reveal that the introduction of alginate causes an increase in crystallinity. As the amount of alginate further increased in PUK-3 to one mole along with the 1,4-BDO (one mole), peak intensity changed to 22,857 a.u. and d-spacing also changed to 3.54 Å. In sample PUK-4, where 1.5 moles of alginate and 0.5 moles of 1,4-BDO were used as a chain extender, d-spacing increased to 3.45 Å, peak intensity increased to 23,606 a.u. and angle 2θ at 25.3 degrees was observed. In sample PUK-5 (prepared with two moles of alginate), d-spacing was 3.66 Å, peak intensity was 31,800 a.u. and angle 2θ was 24.3°. It can be observed from the XRD patterns that, as the alginate contents in the sample increased, crystallinity also increased. This effect might be due to the highly crystalline structure of alginate [32]. Alginate forms a more ordered pattern compared to the amorphous property of 1,4-BDO. Figure 4 shows that the diffraction pattern of the samples changed from amorphous broad shoulder to a relatively higher sharp peak as the chain extender ratio changed.

### 3.4. TGA Analysis

The TGA technique was applied for the analysis of the thermal stability of the synthesized samples. Figure 5 exhibits the degradation pattern of the PUs with different amounts of alginate used as a cross-link. TGA findings including onset temp, T_10_, T_20_, T_30_, T_50_, and T_80_ obtained from thermograms are represented in Table 3. It is apparent from the thermograms that all the samples exhibit three temperature zones for sample degradation [33]. In the first part, a 10% decrease in the weight was observed for all samples in the temperature range from 295 to 365 °C, which might be due to the breakdown of shorter chains [34]. In the middle phase of breakdown, an almost 20% reduction in the weight of the sample occurred in the temperature range from 365 to 425 °C, which might be ascribed to the decomposition of isocyanate and the hydroxyl group. In the last stage, the complex portion of PU composite, e.g., polyol backbone decomposed. It is evident from the results that the introduction of alginate enhances the thermal stability of the sample. In the sample in which only 1,4-BDO was used as a chain extender (PUK-1), it degraded at a low temperature (240 °C). However, samples prepared with alginate (PUK-2 to PUK-5) showed better thermal stability. It is quite obvious from the above discussion that alginate as the cross-link in the chain extension step boosted the heat resistance of samples (PUK-2 to PUK-5) enormously. Thermal stability of PUs were enhanced by chain extenders such as alginate. As the concentration of alginate increased, thermal stability of the PU samples also increased. It can be concluded that alginate was the sole source in enhancing the thermal stability of PU samples.

### 3.5. DSC Analysis of PUs

The thermal transition pattern of the prepared samples was evaluated with the help of differential scanning calorimetry (DSC). Thermograms of PU samples are presented in Figure 6 and Table 3 represents glass transition temperature values (Tg). There are two types of Tg that are obtained after the second heating scan: the Tg acquired below room temperature attributed to the soft segment (SS) and the Tg obtained above room temperature attributed to the hard segment (HS). The sample prepared without alginate (PUK-1) showed a sharp peak at 50 °C that indicated the Tm of the prepared sample [35]. Alginate-based samples (PUK-2 to PUK-5) had a high Tg compared to the neat 1,4-BDO sample PUK-1. It is obvious from the above-mentioned results that the presence of alginate contributes extra flexibility to PU materials and a greater mobility of HS alginate-based PUs compared to neat PU.

### 3.6. Evaluation of Hemolytic Activity

Evaluation of the hemolytic activity of the synthesized PU composite samples was carried out by adopting a method reported in the literature [19,20]. Table 4 shows the hemolytic activities of synthesized materials. According to the results presented in Figure 7 and Table 4, all the prepared sample clearly show % values of the hemolytic activity in an acceptable range below 10%, which is almost negligible and are considered as causing no toxic effect in the human body. The hemolytic studies proved that, as the alginate concentration from PUK-1 (0 mole) to PUK-5 (2 mole) increased in the PU materials, the hemolytic activity shifted to a low range. Alginate is bacteriostatic due to its structural formation that further improves its use in biomedical implants and sutures. These results are in agreement with the conclusion that the synthesized sample does not display toxic effects in the human body after its application. It is obvious that, as the concentration of alginate increased, the toxic effect of PU decreased and biocompatibility improved. The results reveal that, although 1,4-BDO-based PU caused a lesser hemolytic effect, alginate further reduces the hemolytic activity of PU materials, which suggests that alginate-based PUs are more compatible with blood cells [36].

### 3.7. Tensile Strengths of Biodegradable Composites

Mechanical analysis of the synthesized PU samples was carried out and results of tensile strength, hardness, and elongation at break for PU samples are presented in Table 5. It is obvious from the data that tensile strength improves with the addition of alginate contents. As the concentration of alginate increased from PUK-1 to PUK-5 (0 to 2 moles), tensile strength values changed from 7.23 to 9.8 MPa. PUK-5 showed the highest tensile strength as compared to other samples, where lesser alginate content (0.5 to 1.5 mole) were added. Alginate caused reinforcement that resulted in the enhancement of tensile strength and a hardness of resultant composite material, which results in strong interlocking between the PU prepolymer matrix and alginate as the chain extender. The hardness of the PUK-1 sample is 87.20, which is very low compared to PUK-5at 93.12, due to high alginate content in the sample, causing flexibility in polymer chains. Overall, tensile properties improved with the increasing concentration of alginate content.

### 3.8. Contact Angle Measurements

The contact angle for water on the samples was determined and is presented in Table 6. After water was dropped onto the PU films for 30 s, the measured contact angles of PUK-1 to PUK-5 were about 74° to 70°. Water absorption in terms of percentage is also given in Table 6. The results show that, with the increase in alginate concentration, the contact angle decreases, which shows that the hydrophilicity of PUs increases. Sample PUK-1 had a maximum contact angle (74.13°) and sample PUK-5 exhibited inverse behavior with PUK-1, showing a contact angle of 70.10°. The measurement of water absorption shows that, as the amount of alginate increases in PU samples, the percentage of water absorption increases. Both results reveal that the alginate-based samples are more hydrophilic than the PUK-1 sample. This hydrophilicity may be attributed to the ionic groups present in the PU samples, which increase with increases in the alginate amounts in the PU samples.

## 4. Comparative Profile

Alginate is biodegradable, biocompatible, bioactive, less toxic, and is a low cost anionic polysaccharide, and as a part of structural component of bacteria and brown algae (sea weed), it is quite abundant in nature. It is used in combination with polyurethanes to form elastomers, nanocomposites, hydrogels, etc., that especially revolutionized the food and biomedical industries. The current work summarized the development in alginate-based polyurethanes with their potential applications. The results revealed that the structure of the material is elucidated with FTIR and NMR with significant biocompatibility and crystallinity and profound thermal and mechanical properties as compared to the available reported material.

## 5. Conclusions and Future Perspectives

Alginate- and BDO-based PU composites were prepared with different mole ratios. Alginate was converted to smaller units by acid hydrolysis, after which it was blended with 1,4-BDO and was used for the chain extension of PU. The reaction of the biomolecule (ALG) and 1,4-BDO in the PU backbone was confirmed by the ^1^H NMR, FTIR spectroscopy, which found PUs according to the proposed structure. XRD studies showed that the crystallinity of samples increased with an increase in ALG contents. DSC analysis also showed that the glass transition temperature increased with an increase in ALG concentration. It is obvious from the results that alginate also boosted the mechanical properties and thermal stability of the prepared samples. The sample prepared with the highest concentration showed maximum thermal stability. The hemolytic activity values of PU material decreased as the amount of ALG improved in the PUs. Water absorption and contact angle measurement showed that, as the concentration of alginate increased in the samples, the hydrophilicity of samples also increased. Finally, it is concluded that alginate is a suitable chain extender that enhances the characteristic properties of PUs. In addition to the potential array of applications of novel material, certain limitations associated with this unique polymer can be overcome either by modification of their structure or blending with other natural and synthetic polymers. PUs or alginate hydrogels, elastomers, and nanocomposite systems with novelty in their properties make alginates a potent polymer to be explored further.

## Figures and Tables

**Figure 1 polymers-14-02091-f001:**
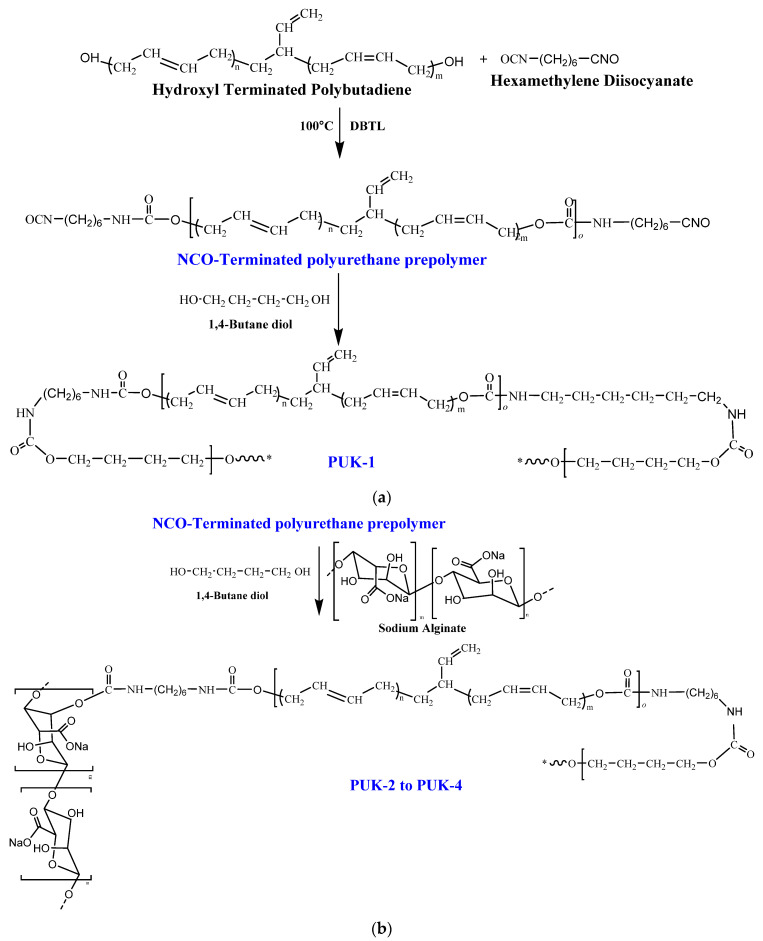
(**a**) Reaction scheme for PUK-1; (**b**) reaction scheme for PUK-2 to PUK-4; (**c**) reaction scheme for PUK-5.

**Figure 2 polymers-14-02091-f002:**
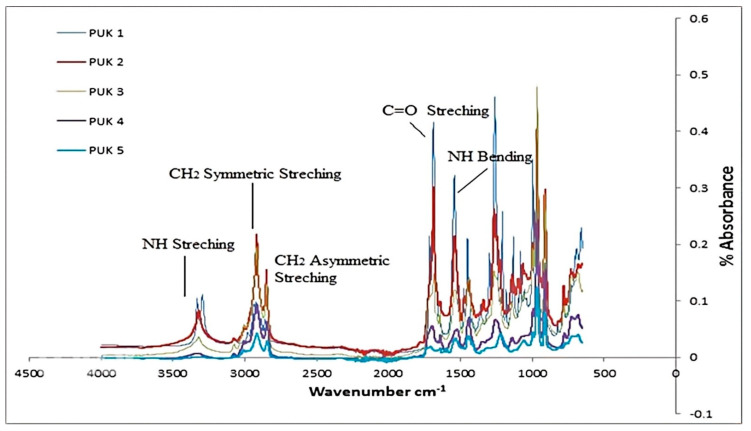
FTIR spectra of alginate- and 1,4-BDO-based PUs.

**Figure 3 polymers-14-02091-f003:**
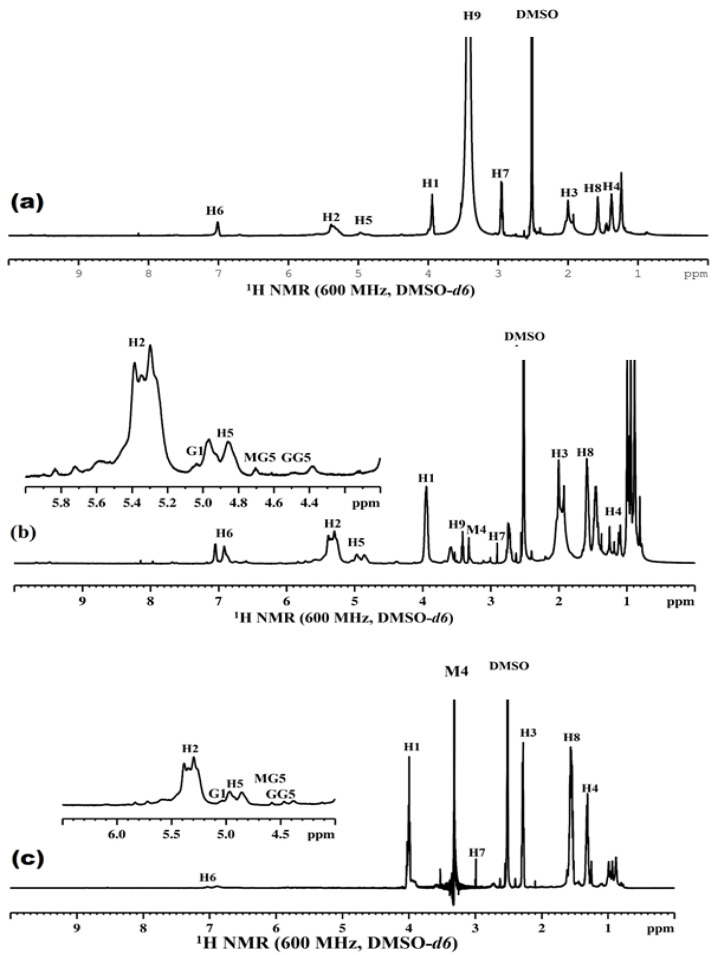
^1^H NMR of (**a**) PUK-1 (**b**) PUK-3 (**c**) PUK-5.

**Figure 4 polymers-14-02091-f004:**
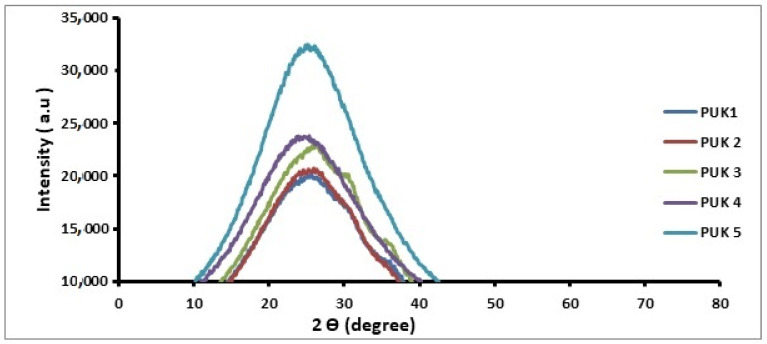
XRD pattern of alginate- and 1,4-BDO-based PUs.

**Figure 5 polymers-14-02091-f005:**
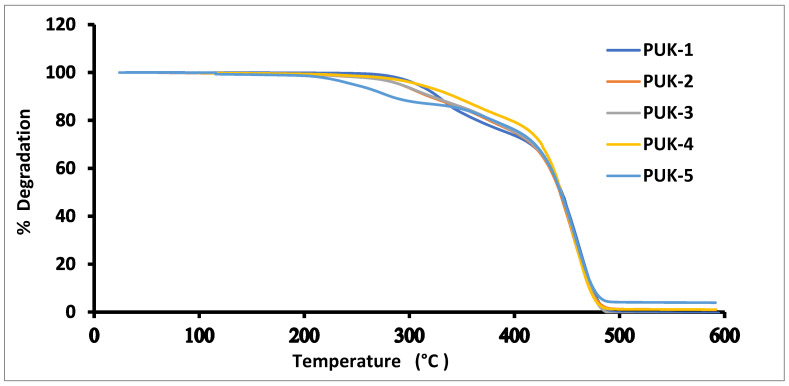
TGA thermograms of alginate- and 1,4-BDO-based PUs.

**Figure 6 polymers-14-02091-f006:**
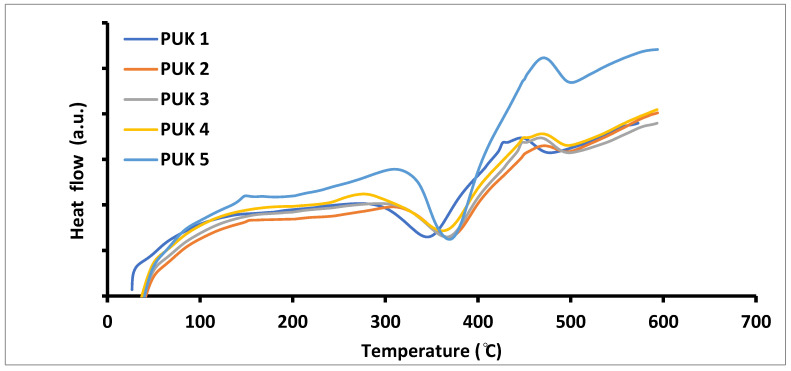
DSC profiles of alginate- and 1,4-BDO-based PUs.

**Figure 7 polymers-14-02091-f007:**
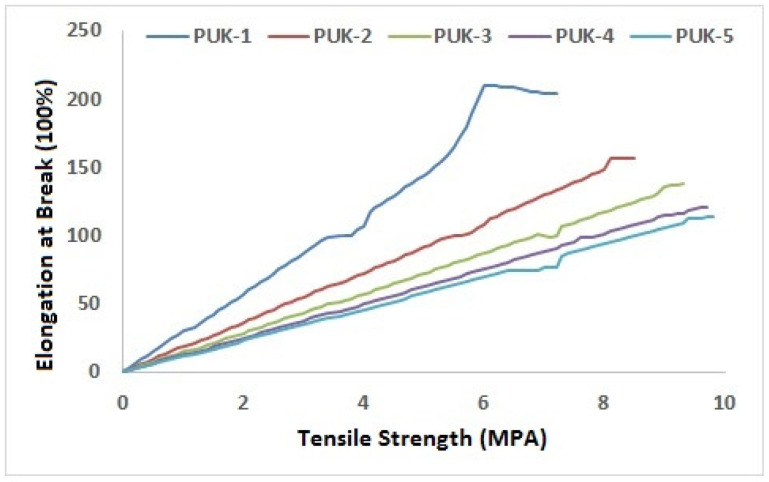
Tensile analysis of PU films.

**Table 1 polymers-14-02091-t001:** Sample code description and PU compositions.

Sr. No.	Sample Code	HTPB ^a^ (Moles)	HMDI ^b^ (Moles)	1,4-BDO ^c^ (Moles)	Alginate
**1**	**PUK-1**	1	3	2	0
**2**	**PUK-2**	1	3	1.5	0.5
**3**	**PUK-3**	1	3	1	1
**4**	**PUK-4**	1	3	0.5	1.5
**5**	**PUK-5**	1	3	0	2

^a^ Hydroxyl-terminated polybutadiene; ^b^ Hexamethylene diisocyanate; ^c^ 1,4- butanediol.

**Table 2 polymers-14-02091-t002:** XRD and DSC analytical profile of polyurethane.

Sr. No.	Sample Code	Chain Extender Ratio Alginate:BDO	2Ɵ (deg)	d-Spacing (Å)	Intensity (a.u)	Glass Transition
**1**	**PUK-1**	0.0:2.0	25.0	3.56	20,197	40.27
**2**	**PUK-2**	0.50:1.50	25.1	3.55	20,641	40.79
**3**	**PUK-3**	1.00:1.00	25.2	3.54	22,857	41.59
**4**	**PUK-4**	1.50:0.50	25.3	3.53	23,606	42.69
**5**	**PUK-5**	2.0:0.0	25.4	3.51	32,347	45.35

**Table 3 polymers-14-02091-t003:** TGA analysis of polyurethane composites.

Sr.	Sample	T_onset_ ^a^	T_i_ ^b^	T_10_ ^c^	T_20_ ^c^	T_50_ ^c^	T_60_ ^c^	T_80_ ^c^	Residue ^d^
#	Code	(°C)	(°C)	(°C)	(°C)	(°C)	(°C)	(%)	(%)
**1**	**PUK-1**	30	260	343	368	393	419	436	1.01
**2**	**PUK-2**	29	270	356	379	400	422	440	3.30
**3**	**PUK-3**	28	275	360	384	405	429	448	5.00
**4**	**PUK-4**	28	273	363	387	408	431	453	6.01
**5**	**PUK-5**	29	254	364	363	387	408	455	8.01

^a^ Temperature from which TGA analysis starts; ^b^ initial decomposition temperature of samples; ^c^ temperatures representing 10%, 20%, 50%, 60%, 80% degradation of samples; ^d^ mass residue percentage at 600 °C.

**Table 4 polymers-14-02091-t004:** Toxicity levels of the prepared polyurethane composites.

Sample Code	% Hemolysis ^a^ (Mean)	Standard Deviation
**PUK-1**	13.9	0.24
**PUK-2**	7.72	0.34
**PUK-3**	6.5	0.15
**PUK-4**	5.4	0.68
**PUK-5**	3.99	0.41
**DMF ^b^**	0.11	0.03
**PBS ^c^**	0.06	0.03
**Triton-X-100**	95.1	0.52

Scale: 1–10, not toxic; 11–25, slightly toxic; 26–50, moderately toxic; 50–100, highly toxic. ^a^ Mean of three values; ^b^ dimethyl formamide; ^c^ phosphate buffer saline.

**Table 5 polymers-14-02091-t005:** Mechanical properties of PU films.

Sr. No.	Sample Code	Tensile Strength (M Pa)	Elongation at Break (%)	Hardness (Shore A)
1	**PUK-1**	7.23 ± 1.05	204 ± 0.85	87.32 ± 0.89
2	**PUK-2**	8.52 ± 1.21	156 ± 0.89	89.52 ± 1.85
3	**PUK-3**	9.34 ± 1.41	138 ± 1.05	90.41 ± 0.87
4	**PUK-4**	9.67 ± 1.01	120 ± 1.15	92.30 ± 1.99
5	**PUK-5**	9.8 ± 1.55	113 ± 1.03	93.12 ± 1.37

**Table 6 polymers-14-02091-t006:** Swelling properties and contact angle measurements of PU films.

Sample	Contact Angle (θ°)	Water Absorption * (%)
**PUK-1**	74.13 ± 2.4	10.69 ± 2.72
**PUK-2**	72.24 ± 3.11	10. 92 ± 3.41
**PUK-3**	71.24 ± 1.85	11.59 ± 2.52
**PUK-4**	70.32 ± 2.63	13.66 ± 3.43
**PUK-5**	70.10 ± 2.31	14.78 ± 2.31

* Absorption is measured after 6 days.

## Data Availability

This study did not report any data.

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
