# Peer review of "Biocompatibility and Hemolytic Activity Studies of Synthesized Alginate-Based Polyurethanes"

_polymers, 2022, doi:10.3390/polym14102091_

Round 1

Reviewer 1 Report

Dear Authors,

I read carefully the paper, is attractive but some issues must be clarified:

  1. In the Introduction part, please modify:

              -a diol is sufficient for polycondensation, not necessary a macrodiol

              -line 35-36 repetition ‘’coating’’

  1. In the case of utilizing Alginate, the functionality is higher than 2, and about Flory-Stockmayer theory a crosslinked product must obtain. How do you measure, in this context de molecular weight????
  2. A preliminary characterization of alginate is necessary.
  3. Please present the novelty of the paper.
  4. In the case of TGA, the 10% decrease assess to the first level, can’t be associated with DMSO evaporation. The boiling point is 189C. Please explain and add a reference.
  5. Please add in the Supplementary Files the mechanical test graphics.
  6. Please check the English.

Author Response

Reviewer # 1:

Correction against the comments of Reviewer # 1 is Green highlighted

Comment 1:   In the Introduction part, please modify:

                        -a diol is sufficient for polycondensation, not necessary a macrodiol

                          -line 35-36 repetition ‘’coating’’

Response:       Thanks for your comment; the correction has been made in macro-diol and coating as suggested.

Comment 2:   In the case of utilizing Alginate, the functionality is higher than 2, and about Flory-Stockmayer theory a crosslinked product must obtain. How do you measure, in this context de molecular weight????

Response:       Thanks for your comment; strongly agree with your statement, that the material might be physically cross-linked and processed for the determination of molecular at my collaborative institute. Therefore, failed to confidently defend this comment, sorry for that.

Comment 3:   A preliminary characterization of alginate is necessary.

Response:       Thanks for your comment; the preliminary information about alginate has been provided in section 2.1.1.

Comment 4:   Please present the novelty of the paper.

Response:       Thanks for your comment; the novelty and strength of this investigation have been clearly stated in the last paragraph of the introduction section. 

Comment 5:   In the case of TGA, the 10% decrease assess to the first level, can’t be associated with DMSO evaporation. The boiling point is 189C. Please explain and add a reference.

Response:       Thanks for your comment; agree with the statement. Correction in the discussion part has been made accordingly.

Comment 6:   Please add in the Supplementary Files the mechanical test graphics.

Response:       Thanks for your comment; the result of the mechanical analysis has been provided as a supplementary file.

Comment 7:   Please check the English.

Response:       Thanks for your comment; the English have been thoroughly checked.

Reviewer 2 Report

Greetings, Editor thank you for providing me with the opportunity to review the article. I reviewed the article with title Biocompatibility and hemolytic activity studies of synthesized  alginate-based polyurethanes.  The article topic is intriguing and promising in the area. Overall, the article structure and content are suitable for the POLYMERS journal. I am pleased to send you major level comments, there are some serious flaws which need to be corrected before publication. Please consider these suggestions as listed below.

  1. The title seems good, but the abstract seems to be wired. Please add one more introductory line of your objective in beginning of abstract.
  2. The formatting of article is very massive, particularly in online submission, the title is wrongly placed. In article, the heading and sub-heading must be bold.
  3. Research gap should be delivered on more clear way with directed necessity for the future research work.
  4. Introduction section must be written on more quality way, i.e., more up-to-date references addressed. Please target the specific gap such as 2015-2021 etc.
  5. The novelty of the work must be clearly addressed and discussed, compare previous research with existing research findings and highlight novelty.
  6. What is the main challenge? Why author choose this material? Please highlight in the introduction part.
  7. Please check the abbreviations of words throughout the article. All should be consistent. For example, what is PU?
  8. Please don’t use lumpy reference (such as: [1-3]). Each reference needs to be properly addressed. Please revise your paper accordingly since same issue occurs on several spots in the paper. Please remove reference 1-3 and simply cite this one article here- Yaqoob, A.A.; Safian, M.T.; Rashid, M.; Parveen, T.; Umar, K.; Ibrahim, M.N.M. Chapter One-Introduction of Smart Polymer Nanocomposites. In Smart Polymer Nanocomposites: Biomedical and Environmental Applications; Elsevier Inc.: Cambridge, MA, USA, 2021; pp. 1–25.
  9. In introduction at the end of Line 38, please cite this article- Safian, M.T.; Umar, K.; Parveen, T.; Yaqoob, A.A.; Ibrahim, M.N.M. Chapter Eight-Biomedical applications of smart polymer composites. In Smart Polymer Nanocomposites: Biomedical and Environmental Applications; Woodhead Publishing Series in Composites Science and Engineering; Elsevier Inc.: Cambridge, MA, USA, 2021; pp. 183–204.
  10. The sentence from Line 39 to 41 seems very weird. Please revise your paper accordingly since same issue occurs on several spots in the paper. The lengthy statement fails to convey the intended message.
  11. The main objective of the work must be written on the more clear and more concise way at the end of introduction section.
  12. Please include all chemical/instrumentation brand name and other important specification.
  13. Please add chemical reagents section and stated all chemical with brand specifications.
  14. Please provide space between number and units. Please revise your paper accordingly since some issue occurs on several spots in the paper.
  15. Regarding the replications, authors confirmed that replications of experiment were carried out. However, these results are not shown in the manuscript, how many replicated were carried out by experiment? Results seem to be related to a unique experiment. Please, clarify whether the results of this document are from a single experiment or from an average resulting from replications. If replicated were carried out, the use of average data is required as well as the standard deviation in the results and figures shown throughout the manuscript. In case of showing only one replicate explain why only one is shown and include the standard deviations.
  16. Please provide high quality image for figure 2,3c and 4.
  17. The caption of figure 4 is weird, why some words are bold. Please revise your paper accordingly since same issue occurs on several spots in the paper.
  18. Each section of the findings requires thorough discussion; a basic explanation is insufficient; please describe each section in a critical way.
  19. Please add a comparative profile section to compare your results and prove how it better than previous.
  20. Section 4 should be renamed by Conclusion and Future perspectives. Conclusion section is missing some perspective related to the future research work, quantify main research findings, highlight relevance of the work with respect to the field aspect.
  21. To avoid grammar and linguistic mistakes, Major level English language should be thoroughly checked. Please revise your paper accordingly since several language issue occurs on several spots in the paper.
  22. Reference formatting need carefully revision. All must be consistent in one formate. Please follow the journal guidelines.

Author Response

Reviewer # 2:

Correction against the comments of Reviewer # 2 is Green highlighted

Comment 1:   The title seems good, but the abstract seems to be wired. Please add one more introductory line of your objective at the beginning of the abstract.

Response:       Thanks for your comment. The introductory line has been added as suggested and highlighted.

Comment 2:   The formatting of the article is very massive, particularly in online submission, the title is wrongly placed. In the article, the heading and sub-heading must be bold.

Response:       Thanks for your comment. The mistakes have been addressed.

Comment 3:   Research gap should be delivered on more clear way with directed necessity for the future research work.

Response:       Thanks for your comment. The research gap and future perspectives have been presented.

Comment 4:   Introduction section must be written on more quality way, i.e., more up-to-date references addressed. Please target the specific gap such as 2015-2021 etc.

Response:       Thanks for your comment. The references have been updated as suggested.

Comment 5:   The novelty of the work must be clearly addressed and discussed, compare previous research with existing research findings and highlight novelty.

Response:       Thanks for your comment. The novelty has been clearly mentioned in the introduction part and highlighted.

Comment 6:   What is the main challenge? Why author choose this material? Please highlight this in the introduction part.

Response:       Thanks for your comment. The Challenges have been stated and highlighted

Comment 7:   Please check the abbreviations of words throughout the article. All should be consistent. For example, what is PU?

Response:       Thanks for your comment. The abbreviations have been provided throughout the manuscript.

Comment 8:   Please don’t use lumpy reference (such as: [1-3]). Each reference needs to be properly addressed. Please revise your paper accordingly since same issue occurs on several spots in the paper. Please remove reference 1-3 and simply cite this one article here- Yaqoob, A.A.; Safian, M.T.; Rashid, M.; Parveen, T.; Umar, K.; Ibrahim, M.N.M. Chapter One-Introduction of Smart Polymer Nanocomposites. In Smart Polymer Nanocomposites: Biomedical and Environmental Applications; Elsevier Inc.: Cambridge, MA, USA, 2021; pp. 1–25.

Response:       Thanks for your comment. The said reference has been added and highlighted.

Comment 9:   In introduction at the end of Line 38, please cite this article- Safian, M.T.; Umar, K.; Parveen, T.; Yaqoob, A.A.; Ibrahim, M.N.M. Chapter Eight-Biomedical applications of smart polymer composites. In Smart Polymer Nanocomposites: Biomedical and Environmental Applications; Woodhead Publishing Series in Composites Science and Engineering; Elsevier Inc.: Cambridge, MA, USA, 2021; pp. 183–204.

Response:       Thanks for your comment. The said reference has been added and highlighted.

Comment 10: The sentence from Line 39 to 41 seems very weird. Please revise your paper accordingly since same issue occurs on several spots in the paper. The lengthy statement fails to convey the intended message.

Response:       Thanks for your comment. The said lines have been reviewed and rephrased.

Comment 11: The main objective of the work must be written on the more clear and more concise way at the end of introduction section.

Response:       Thanks for your comment. The said statement has been provided as suggested and highlighted.

Comment 12: Please include all chemical/instrumentation brand name and other important specification.

Response:       Thanks for your comment. The chemicals and instruments' name has been provided and highlighted.

Comment 13: Please add chemical reagents section and stated all chemical with brand specifications.

Response:       Thanks for your comment. The Chemicals section has been added.

Comment 14: Please provide space between number and units. Please revise your paper accordingly since some issue occurs on several spots in the paper.

Response:       Thanks for your comment. The spaces between numbers and units have been provided throughout the manuscript.

Comment 15: Regarding the replications, authors confirmed that replications of experiment were carried out. However, these results are not shown in the manuscript, how many replicated were carried out by experiment? Results seem to be related to a unique experiment. Please, clarify whether the results of this document are from a single experiment or from an average resulting from replications. If replicated were carried out, the use of average data is required as well as the standard deviation in the results and figures shown throughout the manuscript. In case of showing only one replicate explain why only one is shown and include the standard deviations.

Response:       Thanks for your comment. All the experiments were carried out in 3 replicates. Sorry, the information was not reflected in the Tables by mistakes.  Now added and highlighted.

Comment 16: Please provide high quality image for figure 2,3c and 4.

Response:       Thanks for your comment. The improved quality images have been provided in the revised version.

Comment 17: The caption of figure 4 is weird, why some words are bold. Please revise your paper accordingly since same issue occurs on several spots in the paper.

Response:       Thanks for your comment. The corrections have been made.

Comment 18: Each section of the findings requires thorough discussion; a basic explanation is insufficient; please describe each section in a critical way.

Response:       Thanks for your comment. The required discussion has been provided in all the sections.

Comment 19: Please add a comparative profile section to compare your results and prove how it better than previous.

Response:       Thanks for your comment. The comparative profile section has been provided as suggested and highlighted just before the Conclusion Section.

Comment 20: Section 4 should be renamed by Conclusion and Future perspectives. Conclusion section is missing some perspective related to the future research work, quantify main research findings, highlight relevance of the work with respect to the field aspect.

Response:       Thanks for your comment. Section 4, renumbered as 5 has been revised as “Conclusion and Future perspectives”

Comment 21: To avoid grammar and linguistic mistakes, Major level English language should be thoroughly checked. Please revise your paper accordingly since several language issues occurs on several spots in the paper.

Response:       Thanks for your comment. The English language has been thoroughly checked and revised.

Comment 22: Reference formatting needs carefully revision. All must be consistent in one formate. Please follow the journal guidelines.

Response:       Thanks for your comment. The references have been carefully checked and revised.

Round 2

Reviewer 1 Report

Dear Authors,

  1. In the first format of the manuscript you suggest using an Alginate, extracted from algae, and now provided by Sigma-Aldrich. Please be more concise!!!
  2. Using alginate, you obtain a 3D structure!!! For these products is impossible to make a molecular weight analysis. The 3D structure is insoluble, due to the crosslinking reaction, which supposes to be a chemical bond. In this context please remove from the manuscript the GPC analysis.
  3. Please add the Tensile Analysis to the manuscript.

Author Response

Reviewer # 1:

Correction against the comments of Reviewer # 1 is Yellow highlighted

Comment 1:   In the first format of the manuscript you suggest using an Alginate, extracted from algae, and now provided by Sigma-Aldrich. Please be more concise!!!

Response:       Thanks for your comment; the Alginate, was procured from Sigma-Aldrich, and the sentence has been rephrased in order to make corrections.

Comment 2:   Using alginate, you obtain a 3D structure!!! For these products is impossible to make a molecular weight analysis. The 3D structure is insoluble, due to the crosslinking reaction, which supposes to be a chemical bond. In this context please remove from the manuscript the GPC analysis.

Response:       Thanks for your comment; the said information has been removed from all the relevant sections of the manuscript.

Comment 3:   Please add the Tensile Analysis to the manuscript.

Response:       Thanks for your comment; the Tensile Analysis results have been added as suggested.

Reviewer 2 Report

In the revised manuscript authors carefully considered all of the raised comments and the manuscript is much improved. It can be accepted in the present form.

But the formatting of the article is still not in proper sense, particularly in online submission, the title is wrongly placed. Please check before final publication. 

Author Response

Reviewer # 2:

Correction against the comments of Reviewer # 1 is Green highlighted

Comment 1:   In the revised manuscript authors carefully considered all of the raised comments and the manuscript is much improved. It can be accepted in the present form.

Response:       Thanks for your appreciating comment. Obliged.

Comment 2:   But the formatting of the article is still not in the proper sense, particularly in online submission, the title is wrongly placed. Please check before final publication. 

Response:       Thanks for your comment; the correction in the title has been made by Polymers Editorial Office upon my request.

Round 3

Reviewer 1 Report

Agree